# Curcumin and Resveratrol Regulate Intestinal Bacteria and Alleviate Intestinal Inflammation in Weaned Piglets

**DOI:** 10.3390/molecules24071220

**Published:** 2019-03-28

**Authors:** Zhending Gan, Wenyao Wei, Yi Li, Jiamin Wu, Yongwei Zhao, Lili Zhang, Tian Wang, Xiang Zhong

**Affiliations:** 1Nanjing Agricultural University, Nanjing 210000, China; 2017105067@njau.edu.cn (Z.G.); w_wenyao@163.com (W.W.); 2016105063@njau.edu.cn (Y.L.); 2018105066@njau.edu.cn (J.W.); yanninuist@163.com (Y.Z.); zhanglili@njau.edu.cn (L.Z.); tianwangnjau@163.com (T.W.); 2National Experimental Teaching Demonstration Center of Animal Science, Nanjing 210000, China

**Keywords:** resveratrol, curcumin, intestinal inflammatory, toll-like-receptor 4, intestinal bacteria, weaned piglets

## Abstract

Human infants or piglets are vulnerable to intestinal microbe-caused disorders and inflammation due to their rapidly changing gut microbiota and immaturity of their immune systems at weaning. Resveratrol and curcumin have significant anti-inflammatory, bacteria-regulating and immune-promoting effects. The purpose of this study was to investigate whether dietary supplementation with resveratrol and curcumin can change the intestinal microbiota and alleviate intestinal inflammation induced by weaning in piglets. One hundred eighty piglets weaned at 21 ± 2 d were fed a control diet (CON group) or supplemented diet (300 mg/kg of antibiotics, ANT group; 300 mg/kg of resveratrol and curcumin, respectively, HRC group; 100 mg/kg of resveratrol and curcumin, respectively, LRC group; 300 mg/kg of resveratrol, RES group; 300 mg/kg of curcumin, CUR group) for 28 days. The results showed that compared with the CON group, curcumin alone and antibiotics decreased the copy numbers of *Escherichia coli*. Both curcumin and resveratrol down-regulated the level of Toll-like-receptor 4 mRNA and protein expression in the intestine to inhibit the release of critical inflammation molecules (interleukin-1β, tumor necrosis factor-α), and increase the secretion of immunoglobulin. Our results suggested that curcumin and resveratrol can regulate weaned piglet gut microbiota, down-regulate the TLR4 signaling pathway, alleviate intestinal inflammation, and ultimately increase intestinal immune function.

## 1. Introduction

The intestinal tract is not only the organ of food absorption and digestion, but also the largest host defense organ. The intestine is a microbial ecosystem with a large body of microbes, which contributes to the development and function of the intestinal immune system [1,2]. A multitude of studies have shown that intestinal microbes are closely linked to nutrient absorption and the body’s immune function [3,4,5], and intestinal microbes can directly affect the health of the intestinal mucosa which in turn affects the health of other organs and organisms [6,7]. Therefore, regulating intestinal bacteria is an effective way to alleviate deregulated microbiota damage. Piglets often suffer major stress at weaning. Early weaning in piglets leads to infectious diarrhea, low feed intakes, weight loss, serious intestinal morphology damage, intestinal flora disorder, and intestinal inflammation, due to the change of diet and environment [8,9,10].

Traditionally, antibiotics have been used to relieve weaning stress in piglets, but antibiotic abuse can lead to antibiotic-resistant pathogenic bacteria [11], antibiotic residues and decreased beneficial bacteria [12]. The polyphenolic natural products resveratrol and curcumin have significant antioxidant, anti-inflammatory and immune-promoting effects. Interestingly, previous studies reported that resveratrol and curcumin have anti-bacterial activity, and their effects on bacteria are slightly different [13,14,15,16,17], suggesting that these products may be used in swine diets as natural alternative antibiotics. However, the low bioavailability of curcumin and resveratrol may limit their applications [18,19]. Curcumin has a diphenyl ring structure, and because of its diketone structure, curcumin can undergo keto-enol tautomerization. This keto-enol interconversion imparts curcumin the ability to chelate metals, which allows curcumin to become a hydrogen bond donor or receptor, providing anti-inflammatory, anti-oxidative and anti-apoptotic functions [20]. However, because of this tautomeric nature, curcumin is extremely hydrophobic, which limits its absorption. In addition, poor absorption, fast metabolism, and quick systemic elimination can contribute to the low levels of curcumin in plasma and tissue [19]. To this end, some researchers have found that the combination of curcumin and resveratrol may have mutual effects [21,22,23] to improve curcumin bioavailability. The underlying mechanism of resveratrol in curcumin absorption may be due to improved plasma protein binding [24], since curcumin is extremely hydrophobic, and it should combine with plasma protein such as lipoprotein, hemoglobin, albumin to exert its function. Besides this, resveratrol has a diphenyl ring structure similar to curcumin, which contributed to their mutual effects in improving their bioactivity.

Toll-like receptors (TLRs) have been established to play an essential role in the activation of innate immunity system. TLR signaling pathways are activated by recognizing specific patterns of microbial components [25], subsequently promotes phosphorylation of NF-κB signaling pathway, which can increase pro-inflammatory cytokines secretion [26], leading to gastrointestinal tract injury. Interestingly, resveratrol and curcumin have the ability to down-regulated the expression level of *TLR4* pathway under inflammatory conditions [27,28]. Moreover, inhibition of TLR signaling pathway by resveratrol and curcumin may be associated with regulation of gut microbiota.

Here we investigated effect of curcumin and resveratrol in combination on gut microbiota, intestinal immunity function, and alleviation of intestinal inflammation, and its mechanism. Pigs are very similar to humans in anatomy, genetics and physiology; consequently, pigs are an excellent animal model to study human diseases [27], so, we choose weaned piglets as the animal model in this experiment. Our research may offer guidance for intestinal health and the exploitation of clinical applications.

## 2. Results

The groups considered in this study include (ANT group: 300 mg/kg of antibiotics; CON group: none addition; HRC group: 300 mg/kg of resveratrol and curcumin, respectively; LRC group: 100 mg/kg of resveratrol and curcumin, respectively; RES group: 300 mg/kg of resveratrol; CUR group: 300 mg/kg of curcumin).

### 2.1. The Level of Serum and Intestinal Interleukin and Immunoglobulin

The data are shown in Table 1. Compared with the *CON* group, the level of interleukin-1β (*IL-1β*) were lower (*p* < 0.05) in the HRC, RES and CUR groups from the jejunum and the ANT group from the ileum. The level of tumor necrosis factor α (*TNF-α*) were lower (*p* < 0.05) in the HRC and CUR groups from the jejunum and the HRC and LRC groups from the ileum. The level of Interleukin-10 (*IL-10*) were higher (*p* < 0.05) in the HRC, LRC, RES and CUR groups in the jejunum. Higher (*p* < 0.05) level of immunoglobulin G (IgG) were found in the jejunum, ileum and serum of ANT, HRC, LRC, RES and CUR groups, and higher (*p* < 0.05) level of immunoglobulin A (IgA) were found in the ANT, HRC, LRC groups in the jejunum, ANT, HRC, LRC, RES groups in the ileum and serum, respectively. Compared with the *ANT* group, the level of *IL-1β* and *TNF-α* were lower (*p* < 0.05) in the HRC group and RES groups in the jejunum, higher (*p* < 0.05) *IL-10* level in the HRC, LRC, RES and CUR groups in the jejunum. The HRC group has higher (*p* < 0.05) IgA level in the ileum and serum, higher IgG level in the serum.

### 2.2. Messenger RNA Expression

Compared with the *CON* group, the mRNA expression level of jejunal *IL-1β* was down-regulated (*p* < 0.05) in the HRC, RES and CUR groups (Figure 1), while the HRC, LRC and CUR groups showed decrease of *IL-1β* mRNA in the ileum (*p* < 0.05). 

Up-regulated (p < 0.05) mRNA expression level of jejunal and ileal IL-10 in the jejunum and ileum was observed in the HRC, LRC, RES and CUR groups. The HRC, LRC and CUR groups showed decreased (p < 0.05) mRNA expression level of TNF-α in the jejunum, and the ANT, HRC, RES and CUR groups showed decreased (p < 0.05) mRNA expression levels of TNF-α level in the ileum. The ANT, HRC, LRC, RES and CUR groups decreased (p < 0.05) the mRNA expression level of TLR4 in jejunum compared with that in the CON group, and the ANT, HRC, RES and CUR groups showed decreased (p < 0.05) mRNA expression levels of TLR4 in the ileum while no significant difference we found between each group in TLR2 level in the jejunum and ileum compared with that in the CON group. Compared with the ANT group, the mRNA expression level of IL-10 in the HRC, LRC and RES groups were increased (p < 0.05) in the ileum, IL-1β level were increased (p < 0.05) in the CUR group in the ileum.

### 2.3. The Expression of TLR4 Protein

As the western blot results showed, the expression of TLR4 protein in the ANT, HRC, LRC and CUR groups were decreased (*p* < 0.05) in the jejunum and ileum of weaned piglets compared with that in the CON group (Figure 2). No difference (*p* > 0.05) was found between the ANT group and HRC, LRC, RES, CUR groups.

### 2.4. DGGE Analysis

Different bands represent the 16S rDNA V3 gene fragment of different bacteria in the sample. DGGE electropherogram shows that the band numbers are rich in the jejunum. The microbial flora structure of the ileal chyme is much richer than that of the jejunum, and the microbial species present in the jejunum are also distributed in the ileum (Figure 3). The major difference bands in the ileal chyme were numbered.

The Shannon index (H’), diversity (S) and evenness (E) of the bacteria in each sample were comprehensively analyzed (Table 2). The bacterial diversity index and richness of ileal chyme were higher than those of jejunal chyme. Similar to the DGGE map, the microbial diversity index and richness of the chyme in the ANT group were lower than those in the other groups. In the ileum, the microbial diversity index of the RES group was increased, while decreased in the CUR group compared to the CON group.

### 2.5. DGGE Band Sequencing

We sequenced 10 dominant bands in the DGGE profiles of the ileum digesta, and the relative identifications are reported in Table 3. The sequence similarity of all bands was ≥92% as compared with those available in the GenBank database. *Streptococcus ferus, Enterococcus alcedinis, Enterococcus hirae, Eubacterium sp, Lactobacillus* and some uncultured bacteria were found in the samples. Combine with Figure 3, the band of *Enterococcus, Clostridium* and *Lactobacillus* have changed by curcumin and resveratrol.

### 2.6. Bacteria Copy Numbers and Community

Bacterial copy numbers in the intestinal digesta were affected by resveratrol, curcumin or antibiotics (Figure 4). Compared with the CON group, the copy number of total bacteria were decreased (*p* < 0.05) in the jejunum and ileum of the ANT and CUR groups. The copy number of *E. coli* were decreased (*p* < 0.05) in the ANT, HRC and CUR groups in the jejunum and ileum. In the jejunum, an increased (*p* < 0.05) copy number of *Lactobacillus* were found in the RES group, and the copy number of *Lactobacillus* were increased (*p* < 0.05) in the HRC and RES groups in the ileum but decreased (*p* < 0.05) in the ANT group. Compared with the ANT group, the copy number of *Lactobacillus* were much higher (*p* < 0.05) in all the HRC, LRC, RES and CUR groups in the jejunum and ileum, the copy number of *Bifidobacterium* in the HRC and RES groups were higher than the ANT group. Compared with the CON group, the copy number of *Enterococcus* in the jejunum and ileum were decreased (*p* < 0.05) in the ANT, HRC, LRC groups. The copy number of *clostridium* was decreased (*p* < 0.05) in the ANT and HRC groups in the jejunum and ANT, HRC and RES groups in the ileum.

## 3. Discussion

Curcumin and resveratrol are used to alleviate intestinal infections such as colitis and intestinal inflammation [28,29,30,31] because of their anti-inflammatory ability. Weaned piglets were used to study the effects of resveratrol and curcumin on alleviation of the inflammatory and intestinal microbiota disorder. Our results indicated that the addition of curcumin and resveratrol to the diet has many beneficial effects on weaned pigs, including decreased copy number of harmful bacteria in the intestine, downregulated TLR4 signaling pathways, adjustment of interleukin levels, and increased immunoglobulin levels, which are beneficial for piglets’ growth and development during weaning.

After weaning, the main representative microbiota of piglets changed [32], which may cause gut microbiota disorder, and, impaired bowel function often accompanies weaning. Intestinal mucosal barrier damage and intestinal immune dysfunction, Gram-negative bacteria proliferate in large numbers leading to endotoxins in the blood [33]. In the present study, curcumin and resveratrol could affect intestinal bacteria in weaned piglets. Curcumin exhibited a stronger ability than resveratrol in inhibiting *E. coli* proliferation. On the contrary, resveratrol is better than curcumin in improving proliferation of beneficial bacteria *Lactobacillus* and *Bifidobacterium*. *Escherichia coli.*, a Gram-negative bacterium, is significantly increased after weaning. Studies have shown that different types of *E. coli* can cause multiple types of diarrhea [34], and it is the source of infection for some diseases such as urinary tract infections and early-onset neonatal sepsis [35]. *Lactobacillus* and *Bifidobacterium* are the dominant lactic acid bacterium that plays a major role in human and piglet’s intestine health [36] and commonly found in humans and animal gastrointestinal tract, they are beneficial for health [37]. In addition, as the Shannon index and diversity show, resveratrol or the combined use of curcumin and resveratrol increased the intestinal microbiota diversity compared with the antibiotic group. Improvement of the diversity of gut microbes will help to decrease the micro-ecological environment pressure and enhance the correlation between different bacteria, helping to antagonize the changes in the micro-ecological environment parameters to a certain extent, aiding host health and limiting bacterial pathogen colonization [38]. A previous study [39] showed that, using a radioactive resveratrol dose additional metabolites will detect a high polarity substance, which may be produced by bacterial breakdown products from intestinal degradation. This substance will be combined with dihydroresveratrol and then is absorbed by the intestinal epithelium or excreted by urine. On the other hand, gut microbiota plays an important role in curcumin metabolism and biotransformation, as the microbiota is capable of transforming curcumin formulations, which contain curcumin, demethoxycurcumin, and bisdemethoxycurcumin into a range of catabolites [40]. Therefore, we supposed that resveratrol may improve the absorption of curcumin by improving the gut microbiota diversity.

Curcumin and resveratrol have antimicrobial properties. In the present study, curcumin and resveratrol both exhibited significant inhibiting ability of *Enterococcus* and *Clostridium*, especially in the HRC and CUR groups. Some clinical infections can be due to *Enterococcus* such as urinary tract infections, bacteremia, and bacterial endocarditis [36,41]. Ampicillin, penicillin and vancomycin are commonly used to treat *Enterococci* infections, but *Enterococci* are highly drug-resistant [42]. Some *Enterococci* are intrinsically resistant to β-lactam-based antibiotics as well as various aminoglycosides [42]. Here, our finding may provide an approach to treating *Enterococci* infections using combination of curcumin and resveratrol. In addition, *Clostridium* is one of the major representative genera in piglet intestine. *Clostridium difficile*, a notorious disease, which causes 300,000 to 3,000,000 cases of diarrhea and colitis in the United States every year [43], it is also one of the causes of diarrhea and some diseases. However, there is no direct relationship between resveratrol or curcumin and *Clostridium difficile*, but with *Clostridium* in general. Whether curcumin and resveratrol can be used to treat the *Clostridium difficile* infectious need to further investigate. Inhibition of *Enterococcus*, *Clostridium,* as well as *E. coli* suggests that resveratrol and curcumin can inhibit the proliferation of Gram-negative bacteria. LPS, a membrane component of Gram-negative bacteria [44], will be released into the bloodstream [45,46] when the bacteria die or the cell wall are destroyed, and then the expression of the TLR4 pathway is activated. Therefore, we suppose that resveratrol and curcumin down-regulated TLR4 expression by inhibiting these bacteria.

Many researchers have shown that bacteria are closely related to TLR4 and inflammatory. Klinman et al. [47] reported that bacterial DNA rapidly induces lymphocytes to secrete interleukin and interferon-γ. Heimesaat et al. [48] found that *E. coli* increases and aggravates ileitis development through TLR4-dependent signaling pathway. Increased expression of TLR4 in the epithelium is associated with IBD [49]. Under inflammatory conditions, epithelial TLR4 expression is increased, which contributes to both inflammation as well as immune tolerance [25]. After weaning, a large amount of endotoxins or Gram-negative bacteria enters the blood, TLR4 can be stimulated by these pathogens, which induces the activation of the NF-κB signal pathway and finally results in the release of pro-inflammatory cytokines [50]. In our study, resveratrol, curcumin and their combinations can decrease both the mRNA level and protein expression of TLR4 in weaned piglet intestinal epithelium. HS Youn et al. [50] found that resveratrol suppressed NF-κB activation and cyclooxygenase-2 expression in RAW264.7 cells following TLR3 and TLR4 stimulation, but not TLR2 or TLR9; Fu et al. [51] found that curcumin inhibited the protein expression of TLR4, their findings are consistent with our findings. Activated TLR4 pathway will promote the release of inflammatory factor. Excessive levels of released cytokines can aggravate apoptosis by increasing the expression level of caspase3 [52]. Besides this, TNF-α is a key immune modulator, which can produce free radicals, leading to oxidative stress [53]. Furthermore, inflammatory factors, such as *TNF-α, IL-1β* can cause bad injuries in then intestinal mucosal barrier, leading to tight junction protein loss [50,54,55]. Tight junction proteins maintain a selective permeability pathway. When the tight junction proteins are lost, the permeability of the intestinal mucosal barrier will increase, and gut-derived toxins will be released into the bloodstream, causing chronic low-level inflammation [26]. In addition, excess endotoxin can cause intestinal or other organ apoptosis [56]. The anti-inflammatory ability of curcumin and resveratrol may be associated with inhibition of TLR4 activation in weaning piglets.

## 4. Materials and Methods

### 4.1. Animals and Experimental Design

All the procedures were carried out in accordance with the Chinese Guidelines for Animal Welfare and Experimental Protocol, and were approved by the Institutional Animal Care and Use Committee of Nanjing Agricultural University, China (NJAU-CAST-2015-098). One hundred and eighty (90 ♂ + 90 ♀) with an initial weight of 7.8 ± 0.6 kg hybrid piglets (Duroc × Large White × Landrace, weaned at 28 ± 2 d) were selected. All the pigs were housed in pens in an environmentally controlled room (25.0 ± 0.5 °C), and they were randomly divided into six groups that were fed a control diet supplemented with additives as follows: 159 mg/kg of olaquindox +81 mg/kg of kitasamycin +60 mg/kg of chlortetracycline (ANT); no addition (CON); 300 mg/kg resveratrol and curcumin (HRC); 100 mg/kg resveratrol and curcumin (LRC); 300 mg/kg of resveratrol (RES); 300 mg/kg of curcumin (CUR). All the pigs were allowed to consume feed and water ad libitum. Each treatment has three replicates and ten pigs per replicate. The compositions of the diets are presented in Appendix A and met the NRC (2012) requirements for nutrition. Curcumin (≥98%, product name: CuraGold 95) was obtained as a gift from the Cohoo Bio-tech Research & Development Center; Guangzhou, China, website: http://www.co-hoo.com). Resveratrol (CAS number: 501-36-0, ≥98%) was purchased from Seebio Bio-technology Co. Ltd (Shanghai, China). Curcumin and resveratrol were stored in light-proof containers, and fresh supplementary diets were prepared every two days.

### 4.2. Sample Collection

At 56 d of postnatal age, two piglets were randomly selected in every pen (n = 6). After electrical stunning, blood samples were obtained by jugular venipuncture and centrifuged at 3500 g for 15 min at 4 °C, and then the supernatant was carefully moved to a clean 1.5 mL centrifuge tube and then stored at −80 °C. After blood collecting, piglets were sacrificed by the exsanguination. The entire small intestine starting from the pyloric sphincter to the ileocecal valve was removed from the abdominal cavity and divided into three segments, including the duodenum, jejunum and ileum [57]. Approximately 2 g of the digesta from the middle section of the jejunum and ileum were gently squeezed into sterile tubes and were snap frozen in liquid nitrogen, and then stored at −80 °C for further analysis. Next, the jejunal and ileal segments were immediately washed with ice-cold physiological saline to remove the luminal contents, and the mesenteric attachments were carefully removed. The middle of jejunal and ileal mucosae were scraped using a glass microscope slide. The intestinal mucosae were then snap frozen in liquid nitrogen and then stored at −80 °C. for further analysis.

### 4.3. Serum and Intestinal Cytokines

Mucosa (0.2 g) was weighted and suspended in ice-cold PBS (1.8 mL) homogenized in an ice water bath, then centrifuged at 5000 g for 10 min. The supernatant then collected for the measurement of intestinal factors by an ELISA kit. The levels of interleukin 1 beta (IL-1β), interleukin 10 (IL-10), tumor necrosis factor α (TNF-α), immunoglobulin A (Ig A) and immunoglobulin G (Ig G) in the serum and intestine were determined using ELISA kits for pigs (Elabscience Bio-technology. Co., Ltd, Wuhan, China) according to the manufacturer’s instructions.

### 4.4. Total RNA Extraction and REAL-Time PCR

Total RNA from intestinal mucosa ware extracted using TRIzol reagent (TaKaRa Biotechnology, Dalian, Liaoning, China). The RNA concentration and absorbance at 260 and 280 nm were determined using a NanoDrop ND-2000 UV spectrophotometer (Thermo Fisher, Waltham, MA, USA). After quantification of the concentration, the mRNA was reversed-transcribed into complementary DNA (cDNA) using a reverse transcription kit (TaKaRa Biotechnology) according to the manufacturer’s instructions. The primer sequences are listed in Appendix A and synthesized by Sangon Biotech Co. Ltd. (Shanghai, China). The cDNA was amplified using the ChamQ SYBR qPCR Master Mix (Vazyme Biotechnology, Nanjing, China) in the ABI StepOnePlus™ PCR system, reaction mixture containing 0.4 µL of each forward and reverse primers, 0.4 µL of ROX Reference Dye, 10 µL of SYBR Premix Ex Taq™, 6.8 µL of double distilled H_2_O, and 2 µL of cDNA template. The following thermal profile was used for qRT-PCR: 95 °C for 3 min, followed by 40 cycles of 95 °C for 10 s and 60 °C for 30 s. The relative gene expression was calculated by the 2^−ΔΔCT^ method after normalization to β-actin.

### 4.5. Western Blotting

Proteins were extracted from frozen intestinal mucosae by grinding with RIPA lysis buffer and phenylmethanesulfonyl fluoride (Beyotime Institute of Biotechnology, Nantong, Jiangsu, China). Protein concentrations were measured using a bicinchoninic acid (BCA) kit (Beyotime). Thereafter, 40 μg of protein/lane was electrophoresed in 12% SDS–PAGE gels, followed by transfer to polyvinylidene difuoride membranes and blocking with 5% non-fat dry milk in Tris-buffered saline Tween-20 buffer (0.05% Tween-20, 100 mmol/L Tris–HCl, and 150 mmol/L NaCl, pH 8.0) for 2 h. After blocking, the membranes were incubated overnight with primary antibodies at 4℃. The primary antibodies were Toll-like receptor 4 (1:1000; Abcam; Cambridge, MA, USA) and β-actin (1:5000; Proteintech; Rosemont, IL, USA). The membranes were washed in TBST three times and were processed with an HRP-conjugated secondary antibody (horseradish peroxidase-conjugated anti-mouse IgG, 1:10000; Proteintech) for 60 min at room temperature. The blots were developed using an enhanced chemiluminescence reagents (Merck Millipore, Darmstadt, Germany) followed by autoradiography. Images were recorded using a Luminescent Image Analyzer LAS-4000 system (Fujifilm, Tokyo, Japan) and were quantified by Image-Pro Plus 6.0. β-Actin was used as the internal standard to normalize the signals. 

### 4.6. DNA Isolation and Microbial Quantitative PCR

Intestinal digesta bacterial DNA was extracted using the E.Z.N.A^®^ Stool DNA Kit (Omega Bio-tek, Norcross, Georgia, USA) according to the manufacturer’s instructions. Note that the sample should be reheated in step 4 to obtain Gram-positive bacteria. The DNA concentration and OD 260/280 were determined using NanoDrop ND-2000 UV spectrophotometer. The reaction protocol was as follows: 95 °C for 3 min, followed by 40 cycles of 95 °C for 10 s and 60 °C for 30 s. The reaction mixture (0.4 µL of each of forward and reverse primers, 0.4 µL of ROX Reference Dye, 10 µL of SYBR Premix Ex Taq^TM^, 6.8 µL of doubledistilled H_2_O, and 2 µL of DNA) was amplified using an ABI StepOnePlus^TM^ PCR system, 0.1 μg/μL Bovine Serum Albumin (Solarbio, Beijing, China) were added as recommended in the protocol in real-time PCR reaction mixture. The primers and annealing temperatures was listed in Appendix A [58,59,60]. The primers for total bacteria and *Escherichia coli* were obtained from Chen et al., while *Lactobacillus, Enterococcus, Clostridium* and *Bifidobacterium* designed following Han et al. and Hong et al. [59,61]. Briefly, primers were designed following 16S rRNA downloaded from the GenBank database. To avoid any non-specific amplification, the sequences of all the genera fetched from the database were submitted to DNAStar (MegAlign) program. To ensure that the sequences were complementary pairing with the target genus only, they were checkedwith the GenBank program BLAST. Standard curves were generated using serial dilutions of plasmids obtained by standard PCR using the corresponding primers. The plasmid DNA was diluted to 10-fold serial dilution to construct the standard curves for total bacteria, *Lactobacillus, Escherichia coli, Enterococcus, Clostridium* and *Bifidobacterium*. The results were expressed in log10 gene copy number/g digesta (wet weight).

### 4.7. PCR Amplification and DGGE

PrimersGC338F (5′CGCCCGGGGCGCGCCCCGGGGCGGGGCGGGGGCGCGGGGGGCCT ACG GGA GGC AGC AG -3′) and 518R (5′-ATT ACC GCG GCT GCT GG-3′) were used to amplify V6~V8 regions of 16S rDNA. The reaction mixture comprised 5 μL of 10× PCR buffer (TaKaRa Biotechnology), dNTP (2.5 mM) 3.2 μL (Takara), rTaq (5 U/μL) 0.4 μL (Takara), GC-338F (20 μM) 1 μL, 518R (20 μM) 1 μL, template DNA 50 ng and ddH_2_O up to 50 μL. The amplification program was as follows: predenaturation at 94 °C for 5 min; denaturation at 94 °C for 1 min, renaturation at 55 °C for 45 s, extension at 72 °C for 1 min, 30 cycles; final extension at 72 °C for 10 min. The products were purified and recovered using a DNA Gel Extraction Kit (Omega bio-tek). Electrophoresis was performed in 8% polyacrylamide gels using 35~55% of denaturing gradient in 1× TAE buffer at 56 V and 60 °C for 14 h and finally stained with AgNO_3_ [61].

The DGGE map was quantified using the Quantity one software for the number of bands and strip density of each sample. The Shannon index (H), abundance (S) and equilibrium index (E) were used to compare the diversity of different samples [62].

The DGGE band to be recovered was cut and the target band was recovered using the OMEGA Poly-Gel DNA Extraction Kit. The DNA were re-amplified using the same reaction mixture and amplification program as DGGE PCR amplification. The re-amplified DNA fragment was recovered and purified, ligated into the pMD18-T vector, and transformed into DH5α competent cells, and positive clones were screened for sequence determination. Finally, the sequence was compared to the sequence available in the V6-V8 region of the 16S rDNA sequence of the GenBank DNA database.

### 4.8. Statistical Analysis

Data were expressed as means with SEM (standard error of mean) and analyzed by one-way ANOVA using the SPSS 25.0 software (SPSS, Inc., Chicago, IL, USA). Differences among group means were determined by Duncan’s multiple range test. *p* < 0.05 was considered as statistically significant.

## 5. Conclusions

In conclusion, curcumin and resveratrol could regulate gut microbiota, alleviate inflammatory in weaned piglets and decrease the expression level of the *TLR4* signaling pathway in the jejunum and ileum. Resveratrol and curcumin may inhibit the proliferation of gram-negative bacteria and then down-regulate the expression level of *TLR4*, and finally alleviate intestinal inflammatory and improve intestinal immune function. The beneficial effects of these two polyphenols require further development, and our findings may be helpful in exploit a new health care product and animal production.

## Figures and Tables

**Figure 1 molecules-24-01220-f001:**
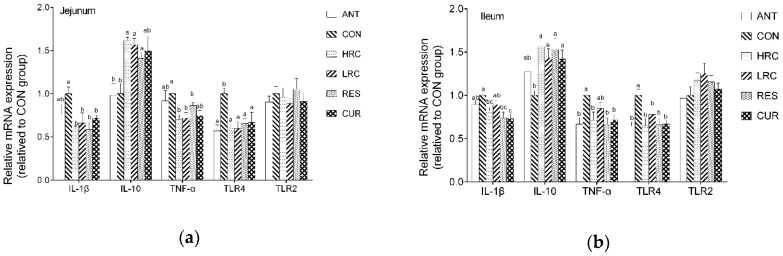
Effects of curcumin and resveratrol supplementation on relative mRNA expression in jejunum (**a**) and ileum epithelium (**b**) of weaned piglets. The column and its bar represented the means value and SEM, n = 6/group. Different letters on the shoulder mark indicate significant difference (*p* < 0.05), the same letter or no letter indicates that the difference is not significant (*p* ≥ 0.05). IL-1β, Interleukin-1β; IL-10, Interleukin-10; TNF-α, Tumor necrosis factor α; TLR4, Toll like receptor 4; TLR2, Toll like receptor 2. ANT, control diet + antibiotics (300 mg/kg); CON, control diet; HRC, control diet + curcumin and resveratrol (300 mg/kg); LRC, control diet + curcumin and resveratrol (100 mg/kg); RES, control diet + resveratrol (300 mg/kg); CUR, control diet + curcumin (300 mg/kg).

**Figure 2 molecules-24-01220-f002:**
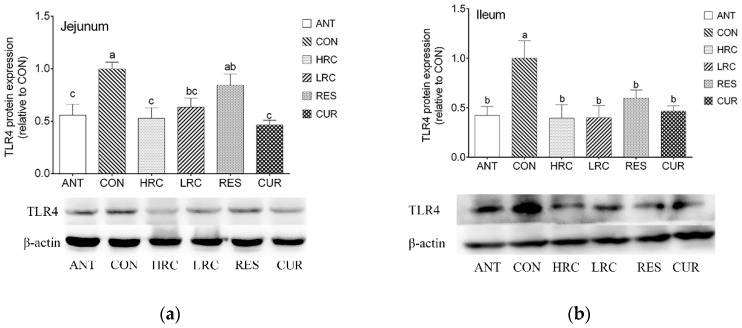
Effects of dietary resveratrol and curcumin supplementation on TLR4 protein in the jejunum (**a**) and ileum (**b**) of weaned piglets. a, ANT group. b, CON group. c, HRC group. d, LRC group. e, RES group. f, CUR group. TLR4, Toll like receptor 4. *ANT*, control diet + 300 mg/kg antibiotics; CON, control diet; HRC, control diet + curcumin and resveratrol (300 mg/kg); LRC, control diet + curcumin and resveratrol (100 mg/kg); RES, control diet + 300 mg/kg resveratrol; *CUR*, control diet + 300 mg/kg curcumin.

**Figure 3 molecules-24-01220-f003:**
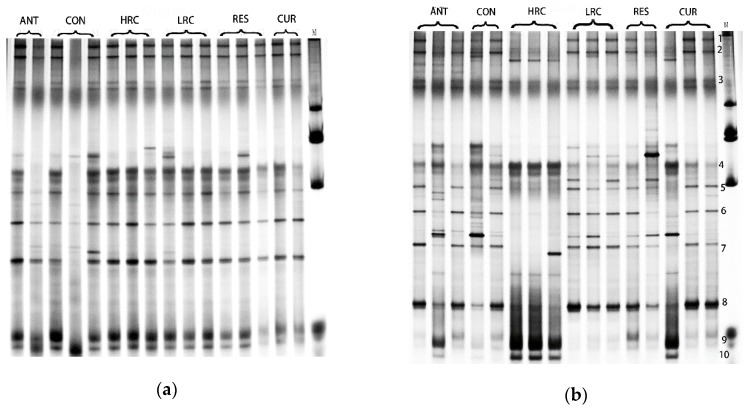
DGGE profiles of V6-V8 regions of 16S rDNA of jejunal digesta (**a**) and ileum digesta (**b**) from weaned piglets. *ANT*, control diet + 300 mg/kg antibiotics; *CON*, control diet; *HRC*, control diet + curcumin and resveratrol (300 mg/kg); *LRC*, control diet + curcumin and resveratrol (100 mg/kg); *RES*, control diet + 300 mg/kg resveratrol; *CUR*, control diet + 300 mg/kg curcumin.

**Figure 4 molecules-24-01220-f004:**
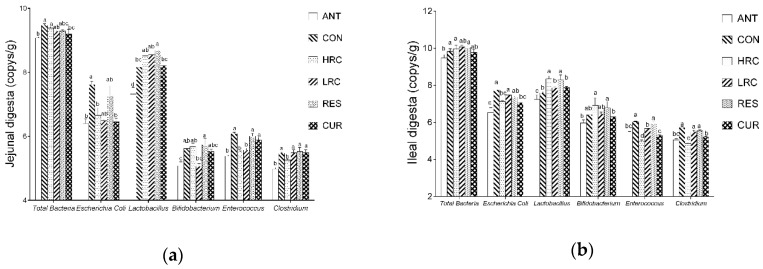
Effects of curcumin and resveratrol supplementation on intestinal bacteria copy number in the (**a**) jejunal and (**b**) ileal. The *column* and its *bar* represented the means value and SEM, n = 6/group. Different letters on the shoulder mark indicate significant difference (*p* < 0.05), the same letter or no letter indicates that the difference is not significant (*p* ≥ 0.05). ANT, control diet + 300 mg/kg antibiotics; CON, control diet; HRC, control diet + curcumin and resveratrol (300 mg/kg); LRC, control diet + curcumin and resveratrol (100 mg/kg); RES, control diet + 300 mg/kg resveratrol; CUR, control diet + 300 mg/kg curcumin.

**Table 1 molecules-24-01220-t001:** Effects of curcumin, resveratrol and antibiotics supplementation on interleukin and immunoglobulin in jejunum, ileum and serum of weaned piglets.

Items	ANT	CON	HRC	LRC	RES	CUR
Jejunum						
IL-1β (ng/g prot)	28.46 ± 3.05 ^ab^	31.53 ± 2.18 ^a^	18.77 ± 1.08 ^c^	25.18 ± 2.71 ^c^	22.69 ± 0.92 ^bc^	23.44 ± 1.13 ^bc^
TNF-α (pg/mg prot)	157.05 ± 7.34 ^a^	167.52 ± 1.96 ^a^	123.82 ± 5.79 ^b^	172.10 ± 4.51 ^a^	121.56 ± 4.16 ^b^	122.51 ± 12.89 ^b^
IL-10 (pg/mg prot)	78.85 ± 2.20 ^c^	88.99 ± 4.03 ^bc^	92.38 ± 2.35 ^ab^	97.57 ± 4.29 ^a^	90.92 ± 3.98 ^ab^	91.21 ± 1.28 ^ab^
IgG (μg/mg prot)	16.10 ± 0.55	15.74 ± 0.69	15.89 ± 0.50	15.84 ± 0.74	15.78 ± 0.37	15.19 ± 0.27
IgA (pg/mg prot)	80.56 ± 2.03	78.98 ± 1.89	79.22 ± 1.57	80.24 ± 2.41	78.15 ± 1.66	76.66 ± 0.77
Ileum						
IL-1β (ng/g prot)	30.33 ± 2.77 ^b^	40.15 ± 2.44 ^a^	31.44 ± 1.06 ^b^	39.84 ± 2.21 ^a^	35.52 ± 1.08 ^ab^	34.49 ± 2.46 ^ab^
TNF-α (pg/mg prot)	163.56 ± 4.81 ^b^	198.63 ± 5.25 ^a^	121.40 ± 8.08 ^c^	131.00 ± 3.39 ^c^	167.21 ± 3.80 ^b^	176.78 ± 7.84 ^b^
IL-10(pg/mg prot)	115.90 ± 4.59 ^ab^	109.47 ± 4.89 ^b^	119.94 ± 4.24 ^ab^	134.31 ± 6.36 ^a^	122.49 ± 9.59 ^ab^	127.26 ± 8.07 ^ab^
IgG (μg/mg prot)	15.25±0.61 ^a^	12.41 ± 0.59 ^b^	16.39 ± 0.57 ^a^	15.66 ± 0.93 ^a^	14.99 ± 0.55 ^a^	15.73 ± 0.50 ^a^
IgA (pg/mg prot)	81.23 ± 2.16 ^a^	74.80 ± 0.95 ^b^	83.24 ± 2.67 ^a^	80.55 ± 2.33 ^a^	74.25 ± 1.27 ^b^	74.25 ± 1.27 ^b^
Serum						
IgA (pg/mL)	25.38 ± 0.2 ^b^	24.53 ± 0.16 ^c^	26.54 ± 0.20 ^a^	25.42 ± 0.26 ^b^	26.10 ± 0.17 ^a^	25.45 ± 0.14 ^b^
IgG (pg/mL)	131.17 ± 1.64 ^bc^	122.21 ± 1.86 ^d^	137.49 ± 0.95 ^a^	129.08 ± 0.98 ^c^	136.43 ± 3.89 ^ab^	128.11 ± 1.42 ^cd^

Data are presented as mean ± SEM, n = 6/group. Different letters on the shoulder mark indicate significant difference (*p* < 0.05), the same letter or no letter indicates that the difference is not significant (*p* ≥ 0.05). *IL-1β*, Interleukin-1β; *IL-10*, Interleukin-10; *TNF-α*, Tumor necrosis factor α; *IgA*, Immunoglobulin A; *IgG*, Immunoglobulin G. *ANT*, control diet + 300 mg/kg antibiotics; *CON*, control diet; *HRC*, control diet + curcumin and resveratrol (300 mg/kg); *LRC*, control diet + curcumin and resveratrol (100 mg/kg); *RES*, control diet + 300 mg/kg resveratrol; *CUR*, control diet + 300 mg/kg curcumin.

**Table 2 molecules-24-01220-t002:** Shannon Index, Evenness, Species richness of intestinal digesta bacteria from DGGE.

Items	H (Shannon Index)	E (Evenness)	S (Species Richness)
Jejunum			
ANT	2.38	0.97	11
CON	2.96	0.97	14
HRC	2.79	0.98	14
LRC	2.60	0.99	14
RES	2.89	0.99	13
CUR	2.73	0.97	13
Ileum			
ANT	2.61	0.97	15
CON	2.86	0.97	20
HRC	2.95	0.98	22
LRC	2.84	0.98	20
RES	3.05	0.96	24
CUR	2.73	0.97	17

*ANT*, control diet + 300 mg/kg antibiotics; CON, control diet; HRC, control diet + curcumin and resveratrol (300 mg/kg); LRC, control diet + curcumin and resveratrol (100 mg/kg); RES, control diet + 300 mg/kg resveratrol; CUR, control diet + 300 mg/kg curcumin.

**Table 3 molecules-24-01220-t003:** Band sequencing in DGGE gels of ileum digesta.

Band Number	Closest Relative Bacteria	NCBI Accession Number	Identity
Band 1	*Nostoc punctiforme*	NR_074317.1	92%
Band 2	*Streptococcus ferus*	NR_115276.1	100%
Band 3	Uncultured bacterium	JX183833.1	100%
Band 4	*Methanocaldococcus bathoardescens*	NR_134839.1	97%
Band 5	*Enterococcus alcedinis*	NR_109727.1	98%
Band 6	*Enterococcus hirae*	NR_114783.2	99%
Band 7	*Clostridium bornimense*	NR_134005.1	99%
Band 8	*Barnesiella viscericola*	NR_121773.2	95%
Band 9	*Chroococcidiopsis thermalis*	NR_102464.1	98%
Band 10	*Lactobacillus johnsonii*	NR_075064.1	97%

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
