# Peer review of "Curcumin and Resveratrol Regulate Intestinal Bacteria and Alleviate Intestinal Inflammation in Weaned Piglets"

_molecules, 2019, doi:10.3390/molecules24071220_

Round 1

Reviewer 1 Report

The manuscript entitled "Curcumin and resveratrol regulate intestinal bacteria and alleviate intestinal inflammatory in weaned piglets" presents some results about the link between intestinal bacteria composition and resveratrol/curcumin enriched diet.

The section of the study about polyphenol and inflammation is coherent but not innovative, the section about intestinal bacteria sounds preliminary and conclusions not fully support by the manuscript data.

The manuscript may potentially become interesting if improved and resubmitted.

Author Response

Point 1. The manuscript entitled "Curcumin and resveratrol regulate intestinal bacteria and alleviate intestinal inflammatory in weaned piglets" presents some results about the link between intestinal bacteria composition and resveratrol/curcumin enriched diet.

The section of the study about polyphenol and inflammation is coherent but not innovative, the section about intestinal bacteria sounds preliminary and conclusions not fully support by the manuscript data.

The manuscript may potentially become interesting if improved and resubmitted.

Response 1. Thank you for your insight comments. We agree that many studies reported polyphenol and inflammation. However, we do not think the mechanism of polyphenol on inflammation or their connection is clear. The question is whether polyphenol alleviates inflammation by regulating gut microbiota. Here we found that curcumin and resveratrol changed the microbial diversity, decreased Enterococcus and clostridium. Importantly, TLRs signal pathway was inhibited and intestinal inflammation was decreased. Since bacteria can activate TLRs signal pathway, therefore, we suppose that curcumin and resveratrol alleviated intestinal inflammation by regulating gut microbiota and inhibiting TLR4 signaling pathway.

We also agree that intestinal bacteria sounds preliminary. Our problem is limitation of resource, including funding and people. We have try our best to revise manuscript in the current stage. 

Reviewer 2 Report

 The manuscript entitled “Curcumin and resveratrol regulate intestinal bacteria and alleviate intestinal inflammatory in weaned piglets” describes effects of two bioactive compounds in the small intestine of pigs. The study included investigations at the level of host gene and protein expression as well as microbial composition, in particular with regard to the immune system. The research might fit into the scope of the journal due to efforts to elucidate the modes of actions of the two bioactive compounds in a pig model.

The points raised in the introduction provide a good overview and interesting aspects for research on resveratrol and curcumin. However, the authors should significantly improve the language of this part. With regard to the orientation of the journal, the chemical properties and the use, which is limited by the chemical instability (curcumin), should be addressed.

The Material and Methods section lacks some important information that needs to be provided:

Since it is not possible to obtain information about the curcumin and resveratrol used from the supplier mentioned, the authors must provide further details. Details and suppliers should also be provided for other mentioned supplements.

At 56d the blood samples were taken, were the piglets slaughtered on that day? Were the piglets fasted; when was the last meal and was it the same for all animals, was this taken into account during slaughter? Please specify.

Please indicate the exact points for intestinal sampling, as this can have a strong influence on the function (L275).

There is no porcine interleukin 10 ELISA kit available from Elabscience. It would be best to give the exact catalogue numbers here. In addition, results presented in the Supplements ‘IL-1ª- for intestinal epithelium.xlsx’ and ‘IL-10 for intestinal epithelium.xlsx’ are both labeled with Interleukin-1β. How were the intestinal samples prepared for analysis by ELISA?

If you provide WB images as a supplement, why not present larger sections to exclude unspecific bindings?

The E.Z.N.A.® Stool DNA Kit protocol provides some optional steps and recommendations (e.g. mechanical disruption and heat incubation). Therefore, ‘according to the manufacturer’s instructions’ seems not to be appropriate. Supplementary table 3: Reference 55 contains only primers for total bacteria; Clostridium primers are not in Ref 56; Lactobacillus primers not matching Ref 55 and 57. Please review.

Line 318: ‘concentration and quantity’ Please revise.

The authors could add some paragraphs to the results section to make it clearer which reference group they refer to.

Figure 2 shows some Western Blot images, but in the supplements several images are provided. What is the difference between the images and which one was selected? Please specify.

Discussion:

Authors should clarify how the suppression of interleukins contribute to be ‘beneficial effects on weaned pigs’.

Clostridium difficile was found neither quantified by PCR nor enriched with DGGE. In the discussion on C. diff it should be clarified that there is no direct relationship between RES and CUR groups and C. diff, but with Clostridium in general.

Authors should provide information on how the dosages of the supplements were selected? In another study they used e.g. 200mg/kg dietary curcumin, while the current diets contained 300mg/kg. Was that chosen arbitrarily?

Due to the scope of the journal, some other questions should be addressed: How stable is resveratrol and curcumin in the feed; which part survives the gastric passage; how high is the intestinal availability?

The term intestinal flora is outdated and no longer appropriate. Please change accordingly.

Author Response

Response to Reviewer Comments

Point 1. The points raised in the introduction provide a good overview and interesting aspects for research on resveratrol and curcumin. However, the authors should significantly improve the language of this part. With regard to the orientation of the journal, the chemical properties and the use, which is limited by the chemical instability (curcumin), should be addressed.

Response 1: Thanks for your suggestion. We also realized the writing of this section is not good. We have revised INTRODUCTION according to your suggestion. We have added why the bioavailability of curcumin is limited, and potential mechanism of combined use of resveratrol and curcumin.

Point 2. Since it is not possible to obtain information about the curcumin and resveratrol used from the supplier mentioned, the authors must provide further details. Details and suppliers should also be provided for other mentioned supplements.

Response 2: Thank you for your suggestion. The resveratrol used in this experiment was purchased from Seebio bio-technology (Shanghai) Co. Ltd. (Lot number: OQ0512A, CAS number: [501-36-0]). Curcumin was obtained from Cohoo Bio-tech Research& Development Center as a gift (Producet name: CuraGold 95; website: http://www.co-hoo.com; Contact Phone: +86 2087467400).  

Point 3. At 56d the blood samples were taken, were the piglets slaughtered on that day? Were the piglets fasted; when was the last meal and was it the same for all animals, was this taken into account during slaughter? Please specify.

Response 3: Thank you for your constructive suggestion. We have modified this part in line 279~282. The piglets were slaughtered at 56 d after blood samples taken. The last meal was taken at 8 pm at 55d and the piglets were fasted for 12 h. The piglets were slaughter at 8 am in 56d. It was the same for all animals.

In order to avoid sample collecting time error, we use parallel sampling method: the first piglet for slaughtered was form the ANT group; the second one was from the CON group; third, the HRC group; fourth, the LRC group; fifth, the RES group; sixth, the CUR group, and repeated this cycle until sampling ending.

Point 4. Please indicate the exact points for intestinal sampling, as this can have a strong influence on the function (L275).

Response 4: We are very sorry for our negligence of instructions sampling. I added digesta collecting in 4.2 sample collection.

The entire small intestine starting from the pyloric sphincter to the ileocecal valve was removed from the abdominal cavity and divided into three segments, including the duodenum, jejunum and ileum. The digesta used in DNA isolation were gently squeezed from the middle of jejunum and ileum, and storaged in sterile cubes in -80 °C for further analysis. The mucosa used in homogenate, RNA extraction, protein extraction was scraped from the middle of jejunum and ileum. Briefly, the jejunal and ileal segments were washed with ice-cold physiological saline to remove the luminal contents, and the mesenteric attachments were carefully removed. The middle of jejunal and ileal mucosae were scraped using a glass microscope slide. The intestinal mucosae were then snap frozen in liquid nitrogen and then stored at -80 °C for further analysis.

Point 5. There is no porcine interleukin 10 ELISA kit available from Elabscience. It would be best to give the exact catalogue numbers here. In addition, results presented in the Supplements ‘IL-1ª- for intestinal epithelium.xlsx’ and ‘IL-10 for intestinal epithelium.xlsx’ are both labeled with Interleukin-1β. How were the intestinal samples prepared for analysis by ELISA?

Response 5: Thank for your correction, and I added samples preparation in the 4.3 serum and intestinal cytokines in line 293~295. Intestinal mucosal scrapings from the jejunum and ileum were collected and stored at -80. 0.2 g of mucosa scrapings were weighted and suspended in 1.8 ml ice-cold PBS and homogenated in ice water bath, and then centrifuged at 5000 g for 10 min. Finally the supernatant was taken for the measurement of intestinal factors by ELISA kit.

The catalogue number as follows: porcine IL-1β ELISA kit (E-EL-P0002c); porcine IL-10 ELISA kit (E-EL-P0005c); porcine TNF-αELISA kit (E -EL-P0010c); porcine IgA ELISA kit (E-EL-P1273c); porcine IgG ELISA kit (E-EL-P0004c). These products have been stopped selling.

Point 6. If you provide WB images as a supplement, why not present larger sections to exclude unspecific bindings? Figure 2 shows some Western Blot images, but in the supplements several images are provided. What is the difference between the images and which one was selected? Please specify.

Response 6. We are sorry for this confusion. Actually, these is original data, not supplement data. We would like to remove these files.

Point 7. The E.Z.N.A.® Stool DNA Kit protocol provides some optional steps and recommendations (e.g. mechanical disruption and heat incubation). Therefore, ‘according to the manufacturer’s instructions’ seems not to be appropriate. Supplementary table 3: Reference 55 contains only primers for total bacteria; Clostridium primers are not in Ref 56; Lactobacillus primers not matching Ref 55 and 57. Please review.

Response 7. Thank for your correction. Actually, we prepared sample using mechanical disruptor, and the sample were incubated at 95 for 5 min followed by incubation at 70 for 15 min.

We modified the primers slightly according to references. The Primers of total bacteria and Escherichia coli were obtained from reference 60, but the primers of Bifidobacterium, Lactobacillus, Enterococcus and Clostridium was designed by ourself following the reference 58 and 60.

Point 8. Line 318: ‘concentration and quantity’ Please revise.

Response 8. Thank you. We changed to ‘concentration and quanlity’

Point 9 The authors could add some paragraphs to the results section to make it clearer which reference group they refer to.

Response 9. Thanks a lot. The group information has been added at the begging of Results section.

Point 10. Authors should clarify how the suppression of interleukins contribute to be ‘beneficial effects on weaned pigs’.

Response 10. Thanks for these valuable suggestions. The beneficial effects of suppression secretion of interleukins such as IL-1β can help alleviate intestinal injury. Inflammatory factors, such as TNF-α, IL-1β can cause severely injury in intestinal mucosal barrier, leading tight junction protein loss (253~258). Excessive level of cytokines release can aggravate apoptotic because increasing the expression level of caspase3. Besides this, TNF-α is a key immune modulator, which can produce free radicals, leading to oxidative stress. (line 251~253)

Point 11. Clostridium difficile was found neither quantified by PCR nor enriched with DGGE. In the discussion on C. diff it should be clarified that there is no direct relationship between RES and CUR groups and C. diff, but with Clostridium in general.

Response 11. Thank you for your correction, we confused the concept of C. difficile and Clostridium. We have already modified this section.

Point 12. Authors should provide information on how the dosages of the supplements were selected? In another study they used e.g. 200mg/kg dietary curcumin, while the current diets contained 300mg/kg. Was that chosen arbitrarily?

Response 12. Thank you for reviewing our article carefully. We are very careful to select the dose of curcumin and resveratrol. Our original idea is that we suppose combination of curcumin and resveratrol have higher bioavailability. Therefore, combination of curcumin and resveratrol at low levels may have similar effects to higher concentrations alone. Many studies and our previous study showed that there are no difference among 200, 300, and 400 mg/kg. That is why 200 mg/kg are used in many studies, because saving cost in animal production. Here we used 100 mg/kg as low concentrations, but 300 mg/kg as higher concentrations.

Point 13. Due to the scope of the journal, some other questions should be addressed: How stable is resveratrol and curcumin in the feed; which part survives the gastric passage; how high is the intestinal availability?

Response 13. Thanks. This is good question! We never tested the stability of resveratrol and curcumin in feed. However, we think it is good for the stability of resveratrol and curcumin in feed, because many antioxidants such as BHA or BHT have been widely used to improve the stability of pharmaceuticals, including vitamins.

After orally ingested curcumin, curcumin and its reduced metabolites are conjugated with glucuronide and sulfate, resulting in curcumin glucuronide, dihydocurcumin glucuronide, tetrahydrocurcumin glucuronoside or monosulfate and mix sulfate/glucuronoside [1]. Using HPLC-MS/MS analysis in the organs, tissues and fluids of piglets, Ortuno et.al found that, after 6 h of intragastric resveratrol administration, resveratrol and its metabolite were detected in intestine, liver, fluids and other organs. In different organs, the metabolites of resveratrol are different. For example, the main metabolite of resveratrol in cecum and colon is dihydroveratrol, which may be related to intestinal microbiota [2].

Reference:

[1] Briskey, D.; Sax, A.; Mallard, A.R.; Rao, A. Increased bioavailability of curcumin using a novel dispersion technology system (LipiSperse). Eur J Nutr 2018, 1-11

[2] Metabolites and tissue distribution of resveratrol in the pig[J]. Molecular Nutrition & Food Research, 2011, 55(8):1154-1168.

Point 14. The term intestinal flora is outdated and no longer appropriate. Please change accordingly

Response 14. Thanks. Done accordingly.

Round 2

Reviewer 1 Report

Authors partially addressed questions raised during the first revision, nontheless, the necessary resources may not be easly accessible.
Priority is still low in absence of a mechanis, authors should at leat speculate about different possibilities.

Author Response

Point 1. Authors partially addressed questions raised during the first revision, nontheless, the necessary resources may not be easly accessible.

Priority is still low in absence of a mechanis, authors should at leat speculate about different possibilities.

Response 1. Thank you for your insight comments. We agree that there are still many problems that still need to be resolved, and we will pay more attention to other possible mechanisms in future research.

Reviewer 2 Report

The authors have significantly improved the manuscript and especially the materials and methods part is now very comprehensible. There are some language corrections that need to be done, e.g.:

Line 22: ‘…numbers of Escherichia coli, both of curcumin…’ to ‘…numbers of Escherichia coli. Both curcumin…’

Line 36: ‘…that, and then,…’ to ‘…, which in turn’

Line 40: ‘dietary’ to ‘diet’

Line 66: ‘LTR’ seems to be ‘TLR’

Line 274: ‘as a gift’ repeated; ‘Product’ instead of ‘Producet’

Line 276: ‘were stored’

Line 373: ‘Conclusions’

Sup. Table 3: ‘Total bacteria’

Line 75: Authors might replace the first part by ‘The groups considered in this study include:’ and should add the CON group.

Author Response

Point 1. The authors have significantly improved the manuscript and especially the materials and methods part is now very comprehensible. There are some language corrections that need to be done

Response 1. Thank you for your professional and meticulous comments, which will improve the quality of this article. We have corrected the error according to your corrections.

Thank you again for your advice in the first and second round review, which is very helpful to us.

Best regards!